# The impact of work values on the professional development of primary and secondary school teachers: A moderated mediation model

**Guoyan Feng**[1], **Keke Shi**[1]*, **Qiyong Huang**[2], **Jingling Ma**[2]

**1** Guangzhou Xinhua University, Guangzhou, China, **2** Headmaster's Office, Zhangpeng Primary School, Ma Chong, Dongguan, China

* skystyle2013@163.com

## Abstract

This study explored the relationship between work values and primary and secondary school teachers' professional development, as well as the mediating role of work engagement and the moderating role of perceived organizational support. Eight hundred and seventy-three primary and secondary school teachers were recruited at random to complete work values scale, work engagement scale, perceived organizational support scale, and professional development of primary and secondary schools teachers scale. The results revealed that work values positively predicted professional development; work engagement partially mediated the relationship between work values and professional development, and with the mediating effect accounted for 54.62% of the total effect; perceived organizational support regulated the direct path and the first half path of this mediation process. This study empirically clarified why (through work engagement) and when (perceived organizational support) work values were positively related to professional development among primary and secondary school teachers' in China, contributing to the existing work values and teachers' professional development literature. Future research could be encouraged to extend the findings by integrating theory of internal motivation with organizational support theory toward deepening current understanding of how and when work values can contribute to a teacher's professional development.

## Introduction

Professional development of teachers has generally been associated with the learning opportunities teachers avail externally, referring to all the planned and unplanned activities which teachers avail to improve their practices [1]. In essence, the professional development of teachers is a process of continuous learning and inquiry that emphasizes the independent development of teachers' professionalism. Under the background of growing demand for diversified and high-quality education as well as insufficient teacher training, many countries face

**Data Availability Statement:** All data files are available from the corresponding author (E-mail: fengguoyan@xhsysu.edu.cn) or Department of Scientific and Development of Guangzhou Xinhua

University (Phone number: 020-87211830; E-mail: gzxhxykyc@163.com; Hyperlink: https://xueke. xhsysu.edu.cn/index.htm).

**Funding:** The author(s) declare financial support was received for the research, authorship, and/or publication of this article. This research was funded by Educational Science Planning Project (Higher Education Project; 2022GXJK383); Guangdong Sports Bureau Project (GDSS2022N143); Guangdong Higher Education Association Private higher education Professional Committee (2022MBGJ073); Guangdong Province Undergraduate College Teaching Quality and Teaching Reform Project (2022J013/2022J038/ Guangdong Education Gao Han2023No. 4); Guangzhou Philosophy and Social Science Planning Project (2021GZGJ164); Dongguang Philosophy and Social Science Planning Project (2024CG85); Guangzhou Xinhua University Science Research Project (2020KYYB07); and Guangzhou Xinhua University College Students Innovation and Entrepreneurship Project (s202313902015/s202213902013); Philosophy and Social Science Program of Guangdong Province, China (GD21YJY04. The funders had no role in study design, data collection and analysis, decision to publish, or preparation of the manuscript.

**Competing interests:** The authors have declared that no competing interests exist

numerous problems and challenges in the professional development of teachers. The research conducted by Wang Shuanglong showed that the motivation of Chinese teachers to participate in professional development was low, only 21.02% of them participating [2]. Of course, many countries have undertaken professional development activities. However, these activities have failed to significantly affect the practices of teachers [3]. One of the reasons behind the failure of the existing professional development programs is that they are externally driven and less informed by the views of teachers themselves [4]. Though frequent calls to design professional development based on this view, experience, and realities (e.g., work values, work environment of teachers or teachers training) of teachers, however, this requirement has remained unattended [5].

In China, the professional development of primary and secondary school teachers should not only meet the expectations of families and local education departments for students' achievement, but also meet needs of the society and the country for all-round development and lifelong learning ability cultivation. Moreover, the professional development of primary and secondary school teachers is crucial for improving the quality of basic education, promoting healthy growth in both basic education and students themselves. Relevant studies have shown that professional development of teachers promotes and enhances internal professionalization through their entire careers [6], which are more likely to create transformative learning opportunities, bringing authentic and long-lasting benefits [7]. It is the source of motivation for teachers to achieve career happiness [8], and also a key factor that affects education reform [9] and prepares them for future educational challenges. Relevant study also revealed that the teacher professional development intervention had positive impacts on both the participating teachers and their middle school students [10]. International experience has shown that high-quality teacher professional development can expand teachers' professional knowledge, improve teaching practice and learning pattern, as well as enhance self-efficacy and job satisfaction [11]. However, the professional development training model for teachers in China is external top-down control model, which puts primary and secondary school teachers in a position of being guided and suppressed in their professional development process, resulting in a lack of autonomy [12].

Recently, relevant studies that focused on the professional development of teachers tried to combine a variety of constructs that included subject matter and pedagogical knowledge, teacher characteristics and personality, teachers' beliefs (e.g., work values), teaching style, teacher appraisal and feedback, class environment and school climate (e.g., organizational support), and other teacher-related variables [13–15]. However, whether work values influences the professional development of primary and secondary school teachers has not been explored in depth. Although researchers have suggested that work values may influence the professional development of primary and secondary school teachers, little empirical testing has been done on this topic in the context of China. Given that the social culture and school environment faced by Chinese primary and secondary school teachers differ from those in other countries, researches are needed to identify factors that can promote their professional development within this specific educational context.

In fact, the main aim of this study is to discuss the relationship between work values and professional development by analyzing the mediating factors, namely work engagement and perceived organizational support. Specific issues that need to be addressed include: (1) Do primary and secondary school teachers' work values affect teachers' professional development? (2) What role do work engagement and perceived organizational support play in the realization of this relationship of influence? The research of this study may not only deepen the understanding of exiting research on the relationship between work values and the professional development of primary and secondary school teachers, but also enrich the theoretical

research foundation and extend the research field of professional development of teachers. Besides, understanding the mechanisms that contribute to professional development is crucial for developing targeted intervention strategies for primary and secondary school teachers in practice. Thus, it is of great significance to explore the promoting factors promoting the professional development of primary and secondary school teachers under this background.

## Theories and hypotheses

**Work values and professional development of primary and secondary school teacher.** Work values, as the core part of individual self-concept, refer to their value orientation, attitude, concept or opinion in the workplace [16]. Essentially speaking, the professional development of teachers is a process of continuous learning and exploration that emphasizes independent growth [17]. With characteristics of persistence and unchangeability, work values play a critical role in guiding and regulating individuals' attitudes and behaviors at work [18]. According to social identity theory, an individual's relevant attributes (e.g., personality traits) and social status (e.g., teachers) can contribute to the formation of their self-definition, while self-definition may influence an individual's cognition (e.g., work values), behavior (e.g., work engagement), and experience in a specific context [19]. This theory also emphasizes that professional identity determines the basic attitude, professional feeling and degree of professional development of individuals. Only with a positive perceived professional identity can individuals continuously seek professional development [20]. Through empirical research, Choi and Jacobs found that teachers with higher self-awareness had a high enthusiasm for participating in professional development-related activities [21]. The research by Adkins and Naumann showed that work values guided individuals to exhibit behaviors consistent with their value cognition at work, resulting in a higher level of performance [22]. Based on the above theoretical support and empirical research results, this study puts forward hypothesis 1.

*H1*: *Work values can significantly positively predict the professional development of primary and secondary school teachers.*

**The mediating role of work engagement.** Previous studies have revealed that work values are an important factor in promoting the professional development of primary and secondary school teachers, however how work values affect the professional development of primary and secondary school teachers remains to be further studied. Work engagement is a positive, fulfilling and work-related mental state, characterized by vitality, dedication and concentration [23]. First, work values have an impact on work engagement. Existing studies have confirmed that the positive factors in work values that motivate the performance of work behavior [24], such as work engagement. The degree of matching between individual work values and organizational values has a significant impact on work engagement [25]. Previous study has revealed that value congruence have a significant positive direct effect on work engagement [26]. At the same time, according to the theory of internal motivation, when facing the professional difficulties and challenges, rural teachers who are full of hope will choose to remain loyal to their professions and maintain a high level of work engagement [27].

Second, work engagement has an impact on the professional development of primary and secondary school teachers. Previous study showed that teachers' individual work engagement had significant positive effects on the improvement of their learning literacy [28]. Another research revealed that the work intensity of primary and secondary school teachers, especially the high degree of work engagement became an important factor restricting the professional development of teachers. The high degree of recessive work engagement was an individual choice led by traditional education in China [29], which contained Chinese characteristics.

Based on the research of the new generation of employees, Wang Qun and other scholars found that individuals with more positive work values were more willing to invest in valuable work, resulting in higher levels of performance [30, 31]. Based on the above analysis, work values may improve the professional development of primary and secondary school teachers by improving the work engagement level of primary and secondary school teachers. Therefore, this study puts forward hypothesis 2.

*H2*: *Work engagement plays an mediating role between work values and the professional development of primary and secondary school teachers.*

**The moderating role of perceived organizational support.**   Perceived organizational support is a comprehensive view that the organization values individuals' contribution and pays attention to his happiness [32], and it is the external motivation mechanism for individual development. According to the concept of resource access development, the resources obtained by individuals from their environment can be converted into their own psychological resources [33]. The more support they get, the more positive psychological state they can show and the more willing they may work hard [34, 35]. Individuals with low perceived organizational support believe that their values and abilities are not recognized, which makes them less willing to participate in work. On the contrary, individuals with high perceived organizational support can reduce their sense of tension and assume more responsibilities and obligations, which will improve the positive impact of individual work values on work engagement [36]. Therefore, perceived organizational support may play a moderating role in the relationship between work values and work engagement. Wen Zhonglin and Ye Baojuan believed that when the first or second half of the mediating path was regulated by the moderating variable, the mediating effect would also be regulated. That is, there will be differences in the mediating effect at different levels of the moderating variable [37]. Compared to individuals with low organizational support, those with high organizational support are more likely to improve their professional development as teachers by increasing their level of work engagement and values.

In addition, some studies have found that when individuals perceive organizational support, they feel an obligation to reciprocate by taking responsibility for their own professional development [2], and work hard to achieve the goals of the organization [32]. Meta-analysis by Rhoades and Eisenberger also confirmed that a high level of organizational support would motivate employees to participate in actions to achieve organizational goals [38], give positive feedback [39], and assume more responsibilities and obligations, thus improving professional development. Therefore, perceived organizational support may regulate the relationship between work values and professional development of primary and secondary school teachers. Based on this, this study puts forward hypothesis 3.

*H3*: *Perceived organizational support not only moderates the direct prediction of work values of professional development, but also moderates the prediction of work values on work engagement.*

In the Job Demands-Resources model, social support is mentioned as one of the resources that an employee can draw upon, in addition to autonomy, control or meaningfulness, to decrease stain from job stressors and increase positive work attitudes such as values, commitment and engagement [40, 41]. This model considers only environmental and individual characteristics of either job demands or resource, therefore most studies group their variables accordingly [42]. However, within the broader framework of Social Exchange Theory, Organizational Support Theory specifically focuses on the importance of perceived organizational support. In addition to specific individuals, the organizations as institutions can be an

alternative source of support in contributing to meaningful work (e.g. promoting professional development) [35].

To sum up, this study builds a moderated mediation model, as shown in Fig 1, which intends to examine the mediating and moderating mechanism that predict the professional development of primary and secondary school teachers based their work values, in order to reveal how and under what circumstances work values affect the professional development of primary and secondary school teachers, and provide empirical support and theoretical reference for promoting the professional development of teachers.

## Methods

### Ethics approval

The study was conducted in accordance with the Declaration of Helsinki, and approved by the ethics committee of the Institutional Review Board at Guangzhou Xinhua University (Document code: 2023L001) in July 14, 2023.

### Participants

In this study, the convenience sampling method was employed to select primary and secondary school teachers in Guangdong Province of China to complete the questionnaires online questionnaire platform Questionnaire Star (www.wjx.cn) from August 1 to October 30, 2023. The criteria for recruiting participants were primary and secondary school teachers who voluntarily wanted to participant should be recruited. The participants were informed of the anonymous submission of the questionnaires, study's purpose, and confidentiality agreement. The participants could refuse or withdraw from the study anytime before submission. A total of 912 questionnaires were obtained in the survey, 39 invalid questionnaires with missed key information, extreme response time or logical questions were eliminated. Finally 873 valid questionnaires were obtained (95.7% efficient). Among them, there are 217 male teachers (24.9%), 656 female teachers (75.1%). The purpose of this study was explained, and all respondents provided informed consent electronically before starting without any payment. Their privacy and anonymity were guaranteed.

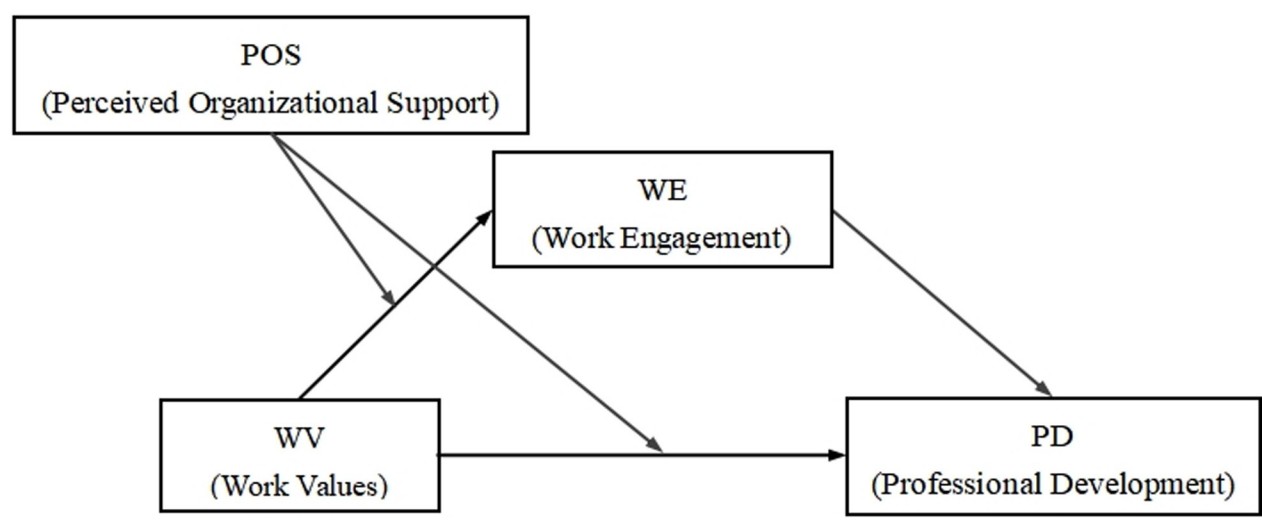

**Fig 1. Proposed moderated mediation model.**

## Measures

***Work values scale.*** The work values scale for primary and secondary school teachers compiled by Xu Xingchun [43], consisted of 27 questions from 7 dimensions: material remuneration, reputation/status, career development, interpersonal relations, organizational management, altruistic dedication and security/stability. The Likert 5-point scoring method was adopted. The higher the score meant that the higher the level of work values of primary and secondary school teachers. ***Professional development of primary and secondly schools scale.*** The scale for professional development of primary and secondary school teachers compiled by Chen Jingjun and others [44] included 21 questions from 5 dimensions: professional sentiment, professional development actions, teacher professionalization concept, professional problem-solving skills, and professional knowledge. The Likert 5-point scoring method was adopted. The higher the score meant that the higher the professional development level of primary and secondary school teachers. ***Work engagement scale.*** The Chinese version of the U-trecht work engagement scale compiled by the Dutch scholar Schaufel [23] and revised by Zhang Yiwen and Gan Yiqun [45] was adopted. The scale had 17 items, including three dimensions of vitality, dedication and concentration. The Likert 5-point scoring method was adopted. The higher the score meant the higher the work engagement t of primary and secondary school teachers. ***Perceived organizational support scale.*** Perceived organizational support scale [46] compiled by Ling Wenyu included 24 items from three dimensions: work support, value identification and interest concern. The Likert 5-point scoring method was adopted. The higher the score, the higher the sense of organizational support for primary and secondary school teachers.

## Control variables

We controlled for demographic characteristics variables that may impact work values, work engagement, perceived organizational support, and professional development of primary and secondary school teachers, including gender, age and teaching age.

## Statistical analysis

SPSS25.0 was used to descriptively count the demographic variables and other variables, and Pearson correlation was used to analyze the four variables of work values, work engagement, perceived organizational support, and professional development of primary and secondary school teachers. It was statistically significant to take $p < 0.05$ as the difference. For all regression analyses, we entered gender, age and teaching age as control variables. All continuous variables were centered to avoid multicollinearity. We used the process macro available for SPSS and SAS (Hayes, 2013) that had become widely used by researchers interested in testing hypotheses about moderation and mediation. The moderated mediation effect test was conducted using the Bootstrap analysis method compiled by Hayes through the Process macro program plug-in, with a significance level of $\alpha = 0.05$ and 95% CI (5000 samples) as the testing criteria.

## Common method biases

The Harman single-factor test method was used to test the homologous deviation. The results showed that 14 factors with eigenvalues greater than 1 were obtained by unrotated principal component factor analysis, and the variance interpreted by the first factor was 14.889%, which was less than the critical standard of 40%. This indicated that there was no serious homologous bias in the research data.

## Results

### Reliability and validity analysis

Work values scale, work engagement scale, professional development of primary and secondary schools teachers scale, and perceived organizational support scale were used in this study. First, the Cronbach's alpha coefficient of work values scale was 0.93, and that for the seven dimensions of this scale ranged from 0.76 to 0.90. The work values variables exhibited good convergent validity, as indicated by the fit indices: $X^2/df = 4.03$, RMSEA = 0.06, and CFI = 0.99. Second, the Cronbach's alpha coefficient of work engagement scale was 0.95, and that for the three dimensions of this scale ranged from 0.86 to 0.88. The work engagement variables exhibited good convergent validity, as indicated by the fit indices: $X^2/df = 3.02$, RMSEA = 0.05, and CFI = 0.99. Third, the Cronbach's alpha coefficient of professional development of primary and secondary schools teachers scale was 0.91, and that for the five dimensions of this scale ranged from 0.83 to 0.93. The CFA was also conducted to test the construct validity of the professional development variables. The fit indices for the CFA model were $X^2/df = 3.76$, RMSEA = 0.06, and CFI = 0.98, suggesting good structural validity. Fourth, the Cronbach's alpha coefficient of perceived organizational support scale was 0.95, and that for the three dimensions of this scale ranged from 0.84 to 0.92. The CFA was also conducted to test the construct validity of the perceived organizational support variables. The fit indices for the CFA model were $X^2/df = 4.72$, RMSEA = 0.07, and CFI = 0.94, suggesting good structural validity. As shown in Table 1.

### Descriptive statistics and correlation analysis

The mean scores for work values, professional development, work engagement and perceived organizational support among primary and secondary school teachers were (M = 3.391, SD = 0.531), (M = 3.906, SD = 0.467), (M = 3.457, SD = 0.700) and (M = 3.064, SD = 0.694). There was a significant positive correlation between work values and work engagement (r = 0.437), perceived organizational support (r = 0.652), and the professional development of primary and secondary school teachers (r = 0.394); and there was a significant positive correlation between work engagement and perceived organizational support (r = 0.500), as well as the professional development of primary and secondary school teachers (r = 0.593); the perceived organizational support was significantly positively correlated with the professional development of primary and secondary school teachers (r = 0.351). As shown in Table 2.

### Mediating effect of work engagement test

Take gender, age and teaching age as control variables to test the mediating effect of work engagement. The results showed that work values had a significant positive predictive effect on the professional development of primary and secondary school teachers ($\beta$ = .362, p < .001),

**Table 1. Cronbach's α coefficient and CFA (X2/df, RMSEA and CFI).**

| Variables | Cronbach's α coefficient | X2/df | RMSEA | CFI |
|---|---|---|---|---|
| 1.Work values | 0.93 | 4.03 | 0.06 | 0.99 |
| 2.Work engagement | 0.95 | 3.02 | 0.05 | 0.99 |
| 3.Professional development | 0.91 | 3.76 | 0.06 | 0.98 |
| 4.Perceived organizational support | 0.95 | 4.72 | 0.07 | 0.94 |

Note. n = 873

**Table 2. Mean, standard deviation and alphas, and inter-correlations.**

| Variables | M | SD | 1 | 2 | 3 | 4 |
|---|---|---|---|---|---|---|
| 1.Work values | 3.391 | 0.531 | 1 | | | |
| 2.Work engagement | 3.457 | 0.700 | .437** | 1 | | |
| 3.Professional development | 3.906 | 0.467 | .394** | .593** | 1 | |
| 4.Perceived organizational support | 3.064 | 0.694 | .652** | .500** | .351** | 1 |

*Note.* $n$ = 873

*$p <$ .05.

**$p <$ .01.

***$p <$ .001.

work values had a significant positive predictive effect on work engagement ($\beta$ = .601, p < .001), and work engagement had a positive predictive effect on the professional development of primary and secondary school teachers ($\beta$ = .329, p < .001). Therefore, work engagement was a mediating variable between work values and the perceived professional development of primary and secondary school teachers. Meanwhile, the direct prediction of work values on the professional development of primary and secondary school teachers was still significant ($\beta$ = .164, p < .001) after adding the mediating variable of work engagement. As shown in Table 3.

The upper and lower limits of the Bootstrap 95% confidence interval for the mediating effect of work engagement and the direct effect of work values on the professional development of primary and secondary school teachers did not contain 0, indicating that work engagement played a partial mediating role between work values and professional development of primary and secondary school teachers. Hypothesis 1 was supported. Among them, the mediating effect accounted for 54.62%. As shown in Table 4.

## Moderating effect of perceived organizational support test

Model 8 in Process compiled by Hayes was adopted to test the mediating effect after controlling gender, age and teaching age. After putting the variable of perceived organizational support into the model and taking the professional development of primary and secondary school

**Table 3. Mediation model testing of work engagement.**

| Projects | Result variables | | | | | | | | |
|---|---|---|---|---|---|---|---|---|---|
| Variables | Professional development | | | Work engagement | | | Professional development | | |
| | $\beta$ | S.E. | $t$值 | $\beta$ | S.E. | $t$值 | $\beta$ | S.E. | $t$值 |
| Gender | -0.020 | 0.034 | -0.570 | -0.038 | 0.050 | -0.757 | -0.007 | 0.030 | -.237 |
| Age | 0.031 | 0.022 | 1.431 | 0.028 | 0.031 | 0.887 | 0.022 | 0.019 | 1.145 |
| Teaching age | 0.041 | 0.016 | 2.496* | 0.083 | 0.024 | 3.469*** | 0.014 | 0.015 | 0.945*** |
| Work values | 0.362 | 0.027 | 13.502*** | 0.601 | 0.039 | 15.409*** | 0.164 | 0.027 | 6.181*** |
| Work engagement | | | | | | | 0.329 | 0.020 | 16.049*** |
| $R^2$ | | 0.447 | | | 0.495 | | | 0.619 | |
| F | | 54.261*** | | | 70.317*** | | | 107.751*** | |

*Note.* $n$ = 873.

*$p <$ .05.

**$p <$ .01.

***$p <$ .001.

**Table 4. Direct, indirect and mediating effects.**

|  | Effect | *Boot S.E.* | *Boot 95% CI* | Proportion |
|---|---|---|---|---|
| Total effect | 0.362 | 0.035 | [0.294; 0.430] |  |
| Direct effect | 0.164 | 0.032 | [0.104; 0.229] | 45.38% |
| Mediating effect | 0.198 | 0.020 | [0.160; 0.236] | 54.62% |

*Note. n* = 873.

teachers as the outcome variable, it was found that work values had a significant positive predictive effect on the professional development of primary and secondary school teachers (β = .171, p < .001), and the product of work values and perceived organizational support had a significant positive predictive effect on the professional development of primary and secondary school teachers (β = .108, p < .001). In addition, taking work engagement as the outcome variable, it is found that work values had a significant positive predictive effect on work engagement (β = .272, p < .001), and the product of work values and perceived organizational support also had a significant positive predictive effect on work engagement (β = .094, p < .05). Hypothesis 2 was supported. Specifically, perceived organizational support not only moderated the direct prediction of work values on professional development, but also moderated the predictive role of work values on work engagement. As shown in Table 5.

Furthermore, according to the method of adding or subtracting one standard deviation from the average, three levels of perceived organizational support were distinguished. The mediating effect of work engagement on work values and perceived professional development of primary and secondary school teachers was analyzed under different perceived organizational support.

The simple slope results showed that for primary and secondary school teachers, work values had a significant positive predictive effect on professional development when perceived organizational support was low (M-1SD) ($\beta$ = .096, $p$ < .01, 95%CI = [0.025; 0.167]), while high perceived organizational support (M+1SD) had a similar effect.

**Table 5. A moderated mediation model testing.**

| Variables | Result variables | | | | | |
|---|---|---|---|---|---|---|
|  | Professional development | | | Work engagement | | |
|  | $\beta$ | S.E. | t | $\beta$ | S.E. | t |
| Gender | 0.004 | 0.030 | 0.131 | -0.002 | 0.048 | -0.031 |
| Age | 0.022 | 0.019 | 1.143 | 0.065 | 0.030 | 2.172* |
| Teaching age | 0.015 | 0.015 | 1.059 | 0.060 | 0.023 | 2.660** |
| Work values | 0.171 | 0.031 | 5.453*** | 0.272 | 0.049 | 5.60*** |
| Work engagement | 0.326 | 0.022 | 15.106*** |  |  |  |
| Perceived organizational support | -0.002 | 0.025 | -0.096 | 0.390 | 0.037 | 10.415*** |
| Work values×Perceived organizational support | 0.108 | 0.025 | 4.209*** | 0.094 | 0.040 | 2.321* |
| $R^2$ |  | 0.396 |  |  | 0.330 |  |
| F |  | 80.965*** |  |  | 71.140*** |  |

*Note. n* = 873.

*p < .05.

**p < .01.

***p < .001.

Work values also had a positive predictive effect on the professional development of primary and secondary school teachers ($\beta$ = .246, $p$ < .001, 95%CI = [0.176; 0.316]). This indicated that the predictive effect of work values on the professional development of primary and secondary school teachers showed an increasing trend with the increase of perceived organizational support. As shown in Table 5 and Fig 2.

For primary and secondary school teachers with low perceived perceived organizational support (M-1SD), work values had a significant positive predictive effect on work engagement ($\beta$ = .067, $p$ < .001, 95% CI = [0.020; 0.113]), while those with high perceived perceived organizational support (M + 1SD), work values also had a positive predictive effect on work engagement ($\beta$ = .110, $p$ < .001, 95%CI = [0.067; 0.158]), and the prediction effect was large. The results indicated that as the perceived perceived organizational support increased, the predictive effect of work values on work engagement tended to increase. As shown in Table 6 and Fig 3.

In addition, at the three levels of low, medium and high perceived organizational support, the mediating effect of work engagement in the relationship between work values and professional development of primary and secondary school teachers showed an increasing trend. This meant that perceived organizational support increased, work engagement was more likely to promote the professional development of primary and secondary school teachers. As shown in Table 6.

## Discussion

This study deeply examined the influence process and mechanism of work values and professional development of primary and secondary school teachers. On the one hand, the study clarified "how work values played a role", that is, to influence the professional development of primary and secondary school teachers through the mediating role of work engagement; on the other hand, it analyzed "when the impact was greater", that is, the direct path and first half of this mediating process were moderated by the perceived organizational support. Specifically, compared to primary and secondary school teachers with low perceived organizational support, those with high perceived organizational support had a stronger positive predictive effect on teachers' professional development and work engagement though their work values. This was consistent with the conclusion of existing studies: the combination of leadership

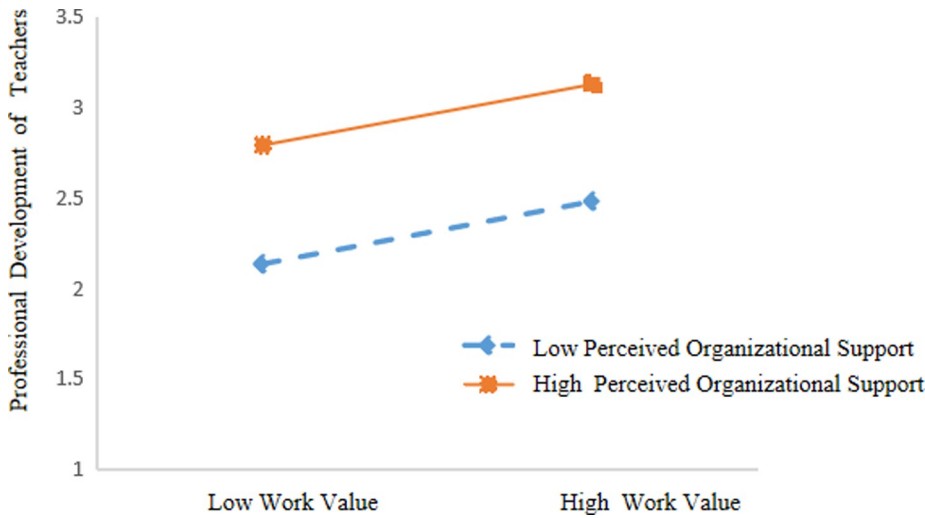

**Fig 2. Perceived organizational support as a moderator between work values and professional development.**

**Table 6. Direct effect and mediating effect from different level of perceived organizational support.**

| | Perceived organizational support | Effect | BootSE | 95% CI |
|---|---|---|---|---|
| Direct effect | 2.37(M-1SD) | 0.096 | 0.036 | [0.025; 0.167] |
| | 3.06 (M) | 0.171 | 0.031 | [0.110; 0.233] |
| | 3.75(M+1SD) | 0.246 | 0.036 | [0.176; 0.316] |
| The mediating effect of work engagement | 2.37(M-1SD) | 0.067 | 0.024 | [0.020; 0.113] |
| | 3.06 (M) | 0.089 | 0.021 | [0.048; 0.131] |
| | 3.75(M+1SD) | 0.110 | 0.023 | [0.067; 0.158] |

support, peer support and teacher work autonomy (e.g. work engagement) in work resources can promote the teachers' full internalization of work values and rules, thus improving teachers' competence [47].

First, through relevant analysis, it was found that work values were positively correlated with the professional development of primary and secondary school teachers, that is, when primary and secondary school teachers demonstrated a higher level of work values in the process of education and teaching, their professional development level may also be enhanced. The finding was in accord with previous research, which showed that individuals with value-oriented work values focused on their learning and growth opportunities, qualifications and achievements in professional fields [48]. The results of the study further revealed the importance of educators paying attention to the professional development of primary and secondary school teachers. Work values were essentially individuals' perception of work and work-related factors, and a clear understanding of individual work values could help managers formulate more effective human resources management policies, thus improving work output.

Therefore, primary and secondary school managers should pay attention to the education of teachers' work values, take altruistic dedication as the spiritual guidance, focus on the education of teachers' professional mission, formulate effective human resources management policies, clarify the content and direction of professional development, and lead primary and secondary school teachers to strive to be good teachers in the new era. Most importantly, the professional development training contents for primary and secondary school teachers should

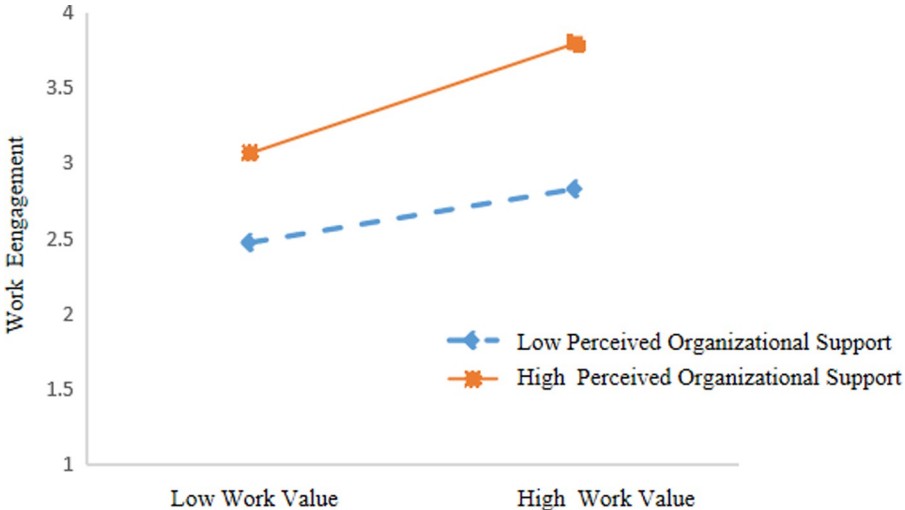

**Fig 3. Perceived organizational support as a moderator between work values and work engagement.**

be strengthened to improve their sense of identity, responsibility, and mission. This would promote the enthusiasm, initiative, and creativity at work [49], establish and improve the scientific work values system, encourage teachers to pursue professional development, and constantly enhance their level of professional development. Teachers should recognize that as a professional, not only responsible for the state and students but also for their professional development [50].

Second, the mediation analysis revealed that work values had a positive effect on work engagement, which in turn was positively associated with the professional development of primary and secondary school teachers. These findings indicated that work engagement partially mediated the relationship between work values and professional development. In accordance with previous research, we found that work values such as establishing a good relationship with colleagues and self-development could lead to positive work outcomes, including individual work dedication [51], which suggested that high-level work values may objectively stimulate the work engagement of primary and secondary school teachers, thereby improving the professional development level of teachers. However, low-level work values were more likely to impede the work engagement of primary and secondary school teachers, which may further hinder their professional development. Furthermore, our findings were consistent with previous studies which demonstrated that work engagement, as a positive and pleasant working state, helped to improve teachers' competence [47]. In other words, teachers with a high level of work engagement were more likely to feel the sense of responsibility and mission as teachers. They were more eager to get a positive response to their hard work in order to improve the professional development of teachers.

Therefore, the results of this study suggested that primary and secondary school teachers should not only pay attention to establishing positive and high-level work values, but also realize the values and responsibility of education and teaching from the bottom of their hearts. They should actively devote themselves to work, which was of great practical significance to professional development of primary and secondary school teachers. Further analysis showed that primary and secondary school teachers with high-level work values were more likely to find the meaning and value in their work and exhibit good professional attitudes, emotions, identity, spirit, and work willingness to work. They were also more likely to enhance their own level of work engagement.

Third, perceived organizational support moderated the direct path and the first half path of this mediation process. Compared to teachers with low perceived organizational support, work values had a more significant direct predictive effect on the professional development of primary and secondary school teachers who had high perceived organizational support. This was consistent with the conclusion of existing studies, which suggested that a combination of leadership support, peer support, and teacher work autonomy as work resources could promote the teachers' full internalization of work values and rules, thus improving teachers' competence [47]. When primary and secondary school teachers perceived high organizations support, they may have a psychological experience of being valued, needed, and respected. This may increase their willingness to work, improve their attitude towards work, enhance trust in and identification with the organization, and make them more likely to establish long-term relationships with colleagues, leaders, and organizations [46]. They were also willing to assume more responsibilities and obligations, so as to improve the professional development of teachers. Compared to teachers with low perceived organizational support, those with high perceived organizational support were more likely to have work values that positively impact their work engagement and improve the professional development of teachers. In other words, when primary and secondary school teachers felt strong and favorable organizational support, they would increase their positive work attitude and work behavior, experience more work

engagement, and achieve better work performance in order to improve the professional development of teachers [52]. High perceived organizational support not only increased individual responsibility and obligation [36], but also enhanced employees' expectations for work and met their needs for respect, recognition and emotional support, and thus repaid the organization with positive work attitude and behavior [38], such as work engagement.

The results of this study suggested that the moderating effect of perceived organizational support should not be ignored when discussing the impact of work values and work engagement on the professional development of primary and secondary school teachers. High perceived organizational support was a contributing factor to the professional development of primary and secondary school teachers. It also enlightened school managers and human resources management departments to formulate corresponding policies, such as financial and non-financial rewards in order to create a supportive school atmosphere and encourage teachers to pursue professional development [2]. Teachers are the foundation of education and the source of teaching. In order to improve the professional development of teachers, primary and secondary schools should provide them with adequate organizational support and guarantee. It is recommended that school managers should take various measures to improve the value level of teachers' work based on the actual situation and work demands of teachers, and encourage teachers to participate creatively [53]. The specific measures can include providing more work support (such as providing promotion opportunities, improving working conditions, etc.) and interest concerns (such as proving special hep for teachers' children get into school, increasing teaching bonuses, etc.), constantly improving the sense of value identity of teachers, so that teachers truly feel from the heart of the school and leaders to the value of their own and work value of the recognition and respect, implementation.

## Limitations and prospect

One limitation of this study was that the convenience sample limits the universality of the results. This study used the questionnaire survey method to explore the relationship between work values, work engagement, perceived organizational support and professional development of primary and secondary school teachers, which was a cross-sectional and not longitudinal study, and could not well infer the causal association between the variables, which may limit the ability to draw causal inferences. Another limitation was that the sample size was limited to primary and secondary school teachers from several cities in Guangdong Province, which may restrict the generalizability of the results to other populations. To improve the research, future studies could include a larger and more diverse sample. Besides, longitudinal follow-up studies or experimental research methods will be carried out, such as using the stochastic intercept cross-hysteresis model (RI-CLPM) to further explore the influence and mechanism of work values on professional development of primary and secondary school teachers, and investigate the longitudinal relationship between work values and professional development in more depth, so as to promote the professional development of primary and secondary school teachers. Moreover, future research could be also encouraged to extend the findings by integrating theory of internal motivation (or cognitive decision theory) with organizational support theory toward deepening current understanding of how and when work values can contribute to a teacher's professional development.

## Conclusions

In this study, it was found that primary and secondary school teachers in China with high work values tended to have increased professional development through work engagement, especially those with a high perceived organizational support. The results indicated that

perceived organizational support acted as a moderator between work values, work engagement, and professional development of primary and secondary school teachers. This research shed light on the antecedent factors of professional development and demonstrates that enhancing teachers' perceived organizational support could lead to improvements in professional development. Overall, this study not only contributed to the existing literature by highlighting the importance of work engagement and perceived organizational support in the professional development among teachers with high work values, but also had important theoretical significance and practical value for improving the professional development of primary and secondary school teachers.

## Supporting information

**S1 Dataset. Dataset used for analyses in present study.**
(SAV)

**S1 File. STROBE statement.**
(DOCX)

## Author Contributions

**Conceptualization:** Guoyan Feng, Keke Shi, Qiyong Huang.

**Data curation:** Guoyan Feng, Keke Shi, Qiyong Huang.

**Formal analysis:** Guoyan Feng, Keke Shi.

**Investigation:** Guoyan Feng, Keke Shi, Qiyong Huang, Jingling Ma.

**Methodology:** Guoyan Feng, Keke Shi.

**Resources:** Guoyan Feng.

**Writing – original draft:** Guoyan Feng, Keke Shi, Qiyong Huang, Jingling Ma.

**Writing – review & editing:** Guoyan Feng, Keke Shi, Jingling Ma.

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
