## [Decision Letter · Decision Letter 0]

22 Mar 2024

PONE-D-24-02012The impact of work values on the professional development of primary and secondary school teachers: A moderated mediation modelPLOS ONE

Dear Dr. Feng,

Thank you for submitting your manuscript to PLOS ONE. After careful consideration, we feel that it has merit but does not fully meet PLOS ONE’s publication criteria as it currently stands. Therefore, we invite you to submit a revised version of the manuscript that addresses the points raised during the review process.

We look forward to receiving your revised manuscript.

Kind regards,

Zahra Masood Bhutta

Academic Editor

PLOS ONE

Journal Requirements:

The author(s) declare financial support was received for the research, authorship, and/or publication of this article. This research was funded by Educational Science Planning Project (Higher Education Project; 2022GXJK383); Guangdong Sports Bureau Project (GDSS2022N143); Guangdong Higher Education Association Private higher education Professional Committee (2022MBGJ073); Guangdong Province Undergraduate College Teaching Quality and Teaching Reform Project (2022J013/2022J038); Guangzhou Philosophy and Social Science Planning Project (2021GZGJ164); Guangzhou Xinhua University Science Research Project (2020KYYB07); and Guangzhou Xinhua University College Students Innovation and Entrepreneurship Project (S202313902015)

3. In the online submission form, you indicated that The original contributions presented in the study are included in the article/supplementary material, further inquiries can be directed to the corresponding author(fenggy3@mail2.sysu.edu.cn).

5. One of the noted authors is a group or consortium. In addition to naming the author group, please list the individual authors and affiliations within this group in the acknowledgments section of your manuscript. Please also indicate clearly a lead author for this group along with a contact email address.

6. Please remove your figures from within your manuscript file, leaving only the individual TIFF/EPS image files, uploaded separately. These will be automatically included in the reviewers’ PDF.

Reviewers' comments:

Reviewer's Responses to Questions

**Comments to the Author**

1. Is the manuscript technically sound, and do the data support the conclusions?

Reviewer #1: Yes

Reviewer #2: Partly

2. Has the statistical analysis been performed appropriately and rigorously? 

Reviewer #1: Yes

Reviewer #2: Yes

3. Have the authors made all data underlying the findings in their manuscript fully available?

Reviewer #1: No

Reviewer #2: Yes

4. Is the manuscript presented in an intelligible fashion and written in standard English?

Reviewer #1: Yes

Reviewer #2: Yes

5. Review Comments to the Author

**Reviewer #1:** Comments to Authors

Abstract

Overall, the abstract provides a concise overview of the study's objectives, methodology, key findings, and implications. Here are some critical comments for the authors:

1. Regarding the structure and alignment, the abstract would benefit from a more organized layout to enhance readability and coherence.

2. Ensuring that each sentence flows logically into the next and that key points are clearly emphasized could improve the overall structure.

3. Additionally, the abstract is somewhat lengthy and could be condensed to 200 words or less to maintain brevity while still effectively summarizing the study's objectives, methodology, findings, and implications.

4. Finally, while the abstract is generally well-written, attention to minor language and grammar issues, such as awkward phrasing or punctuation errors, is recommended to enhance clarity and professionalism. Addressing these aspects would help strengthen the abstract and make it more impactful for readers.

Introduction

1. The introduction is indeed well-written, providing a comprehensive overview of the research topic and its significance. However, there are a couple of areas where the authors could improve:

2. While the content is strong, attention to layout details such as spacing and alignment would enhance the presentation quality. Ensuring consistent formatting throughout the text will contribute to a polished appearance.

3. The authors should review the reference style to align with the guidelines of PLOS ONE. Consistency in referencing style is crucial for professionalism and adherence to journal requirements.

4. Although the introduction effectively identifies the gap in the literature, the specific research objectives could be articulated more clearly to guide readers on what to expect from the study. Clearly stating the research questions or hypotheses would provide readers with a roadmap for understanding the study's aims and contributions.

5. While the conclusion of the introduction emphasizes the significance of the study, the authors could further elaborate on the unique contributions of the research and how it advances knowledge in the field. Providing a more detailed discussion of the potential theoretical and practical implications of the study's findings would enhance the clarity and impact of the introduction.

Literature Review

1. Ensure that the referencing style aligns with the guidelines of the target journal, such as PLOS ONE. Consistency in referencing style is crucial for maintaining professionalism and meeting journal requirements.

2. Incorporate more updated references to ensure the literature review reflects the most current research in the field. Including recent studies will enhance the relevance and credibility of the literature review.

3. Clarify and separate the hypotheses related to the direct relationship between the independent variable (work values) and the mediator (work engagement), as well as the moderation effects of perceived organizational support. Presenting these hypotheses separately would improve clarity and facilitate understanding for the reader.

4. Provide a brief theoretical background before presenting the research model. This background should succinctly summarize relevant theoretical frameworks or concepts that support the study's hypotheses and research model. This addition will enhance the theoretical foundation of the study and provide context for the research model.

Methodology

1. The methodology section appears to prioritize demographic details over discussing the actual study methodology. It is recommended to provide a brief discussion of the demographic characteristics within the text and present the detailed demographic information in a table for better readability and organization.

2. The methodology does not address how data were collected from multiple provinces. If the data were collected online, it raises questions about the high response rate of 95.7%, which is unusually high for online surveys. Authors should provide justification for this high response rate to ensure the reliability and validity of the collected data.

3. The methodology discusses the measurement scales in detail. However, it would be more effective to consolidate this information into a single paragraph for clarity. Additionally, the discussion of reliability measures such as Cronbach's alpha and confirmatory factor analysis (CFA) should be moved to the results section, and the results should be presented in tabular format for easier reference.

4. As the authors is using control variables in the study also, I would suggest to provide brief detail about your control variables along with justification for using these variables as control.

5. The track changes should be removed from the manuscript, particularly from lines 191 to 193, to ensure the document's professional presentation.

Results

1. The results section is commendably well-written and presented, demonstrating a strong presentation of findings. However, one aspect that requires attention is the discussion of scale reliability and validity. It is essential to include a detailed discussion of these aspects within the text of the results section, along with presenting the corresponding statistics in a table format. This will ensure a comprehensive evaluation of the measurement instruments used in the study and enhance the transparency of the findings.

Discussion

2. The discussion is again well written however I would suggest, Provide a more thorough integration of current results with previous research on work values, work engagement, and organizational support in the context of teacher professional development.

3. Elaborate on specific strategies or interventions based on the findings and suggest future research directions to address remaining questions or potential limitations.

Language Quality

1. The paper is generally well-written and clear in its presentation. I observed few complex and long sentences which is suggested simplifying complex sentences to improve papers quality.

2. A thorough proofreading for grammatical accuracy and consistency in terminology would enhance the paper's overall quality.

Recommendation

I recommend accepting this paper with minor revisions as outlined above.

**Reviewer #2:** The “Literature Review” part should include more latest literatures. The “Methods” part is well-written. The methods are employed appropriately and able to meet the objectives of the study. The “Results” part is well-written. The conclusion adequately ties together the other elements of the paper. The study should include “Policy Implications”.

6. PLOS authors have the option to publish the peer review history of their article (what does this mean?). If published, this will include your full peer review and any attached files.

Reviewer #1: **Yes: **AMIN UR RAHMAN

Reviewer #2: **Yes: **Sayed Farrukh Ahmed

---

## [Author Response · Author response to Decision Letter 0]

18 Jun 2024

Response to Reviewers

Dear Reviewer(s),

Thank you very much for taking the time to carefully review and evaluate our manuscript. We will seriously consider your suggestions and incorporate them comprehensively when revising the manuscript, and we will make every effort to enhance the paper and ensure its academic merit. Once again, we sincerely appreciate your attention to our research and your valuable input.

Abstract

1.Regarding the structure and alignment, the abstract would benefit from a more organized layout to enhance readability and coherence. 

My response: Thank you for your suggestion. We have seriously learned other abstracts of research articles published in PLOS ONE, and we agree with you that the abstract can benefit from a more organized layout. Therefore, we have adjusted the structure and alignment of the abstract. You can see changes in the “Abstract”.

2.Ensuring that each sentence flows logically into the next and that key points are clearly emphasized could improve the overall structure. 

My response: The overall structure are not changed, however we have emphasized the key points (the minutiae were deleted) and ensured that each sentence flows logically into the next. You can see changes in the “Abstract”.

3.Additionally, the abstract is somewhat lengthy and could be condensed to 200 words or less to maintain brevity while still effectively summarizing the study's objectives, methodology, findings, and implications. 

My response: We have noticed that the abstract is somewhat lengthy. In order to maintain brevity the contents of the abstract were condensed to less 200 words. However, the study’s objectives, methodology, findings, and implications were still effectively summarized. You can also see changes in the “Abstract”.

4.Finally, while the abstract is generally well-written, attention to minor language and grammar issues, such as awkward phrasing or punctuation errors, is recommended to enhance clarity and professionalism. Addressing these aspects would help strengthen the abstract and make it more impactful for readers.

My response: Thank you for your high comments on my abstract and your careful proofreading. We are so sorry that there are still minor language and grammar issues, such as teachers’ (the correct is teachers’ and the whole paper were changed by teachers’, lines 35/36/39 and so on) and clarifies (the correct is clarified, line 33). In the following days, We will pay more attention to the language and grammar issues to enhance clarity and professionalism. Thanks.

Introduction

1.The introduction is indeed well-written, providing a comprehensive overview of the research topic and its significance. However, there are a couple of areas where the authors could improve:

2.While the content is strong, attention to layout details such as spacing and alignment would enhance the presentation quality. Ensuring consistent formatting throughout the text will contribute to a polished appearance.

My response: Thank you for your high comments on my introduction and your attentive proofreading. According to the submission guidelines of PLOS ONE, the manuscript text have been double-spaced, and the alignment have been left justified. The layout and spacing have met specifications and requirements of PLOS ONE. The consistent formatting throughout the paper become better than before.

3.The authors should review the reference style to align with the guidelines of PLOS ONE. Consistency in referencing style is crucial for professionalism and adherence to journal requirements.

My response: We appreciate your attention to detail and we apologize for any errors or inaccuracies in the references list. We have carefully reviewed the references and made the necessary corrections to ensure accuracy. You can see changes in the “References”.

4.Although the introduction effectively identifies the gap in the literature, the specific research objectives could be articulated more clearly to guide readers on what to expect from the study. Clearly stating the research questions or hypotheses would provide readers with a roadmap for understanding the study's aims and contributions.

My response: Thanks for your advice. In the last paragraph of the Introduction, we have mentioned the main aim (discussing the relationship between work values and professional development by analyzing the mediating factors, namely work engagement and perceived organizational support--the specific questions include: (1)Do primary and secondary school teachers’ work values affect teachers’ professional development？(2)What role do work engagement and perceived organizational support play in the realization of this relationship of influence.) and the contributions(contributing to existing literature on school culture and environment factors in the theoretical model of the professional development and devising policy makers about the importance of work environment and its impact on teachers' professional development and engagement, providing practical guidance to promote the professional development of teachers from primary and secondary school) of the study, which can be articulated to guide readers on what they expect from this study. In addition, the research questions and hypotheses have been provided in the next paragraph/part “Theories and Hypotheses”, the readers can also get them easily, because we marked them with “H1:/H2:/H3:” and provided a great deal of evidences.

You can see the changes from lines 88 to 91.

5.While the conclusion of the introduction emphasizes the significance of the study, the authors could further elaborate on the unique contributions of the research and how it advances knowledge in the field. Providing a more detailed discussion of the potential theoretical and practical implications of the study's findings would enhance the clarity and impact of the introduction.

My response: Thank you for your affirmation and encouragement to the conclusion of the introduction in this article. Under your suggestion we have further elaborated on the unique contributions of the research, including theoretical and practical significance (the added contents are as follows: The research of this study may not only deepen the understanding of exiting research on the relationship between work values and the professional development of primary and secondary school teachers, but also enrich the theoretical research foundation and extend the research field of professional development of teachers. Besides, understanding the mechanisms that contribute to professional development is crucial for developing targeted intervention strategies for primary and secondary school teachers in practice.). See the changes from lines 91 to 96. 

Just as you say, we are totally agree with you that providing a more detailed discussion of the potential theoretical and practical implications of the study’s findings would enhance the clarity and impact of the introduction. We have learned and got it.

Literature Review

1.Ensure that the referencing style aligns with the guidelines of the target journal, such as PLOS ONE. Consistency in referencing style is crucial for maintaining professionalism and meeting journal requirements.

My response: We appreciate your attention to detail and we apologize for any errors or inaccuracies in the references list. We have carefully reviewed the references and made the necessary corrections to ensure accuracy. Vancouver style is used in this paper. 

You can see the changes in the “References” from lines 527 to 691.

2.Incorporate more updated references to ensure the literature review reflects the most current research in the field. Including recent studies will enhance the relevance and credibility of the literature review.

My response: We have added/incorporated the updated references to ensure the literature review reflects the most current research and enhance the relevance and credibility of the literature review in this field. You can see the changes in [7][10][12][26][28][29][35][40][42]

[47], which have been marked with yellow color in the paper.

3.Clarify and separate the hypotheses related to the direct relationship between the independent variable (work values) and the mediator (work engagement), as well as the moderation effects of perceived organizational support. Presenting these hypotheses separately would improve clarity and facilitate understanding for the reader.

My response: We have clarified and separated the three hypotheses: Hypothesis 1(H1): Work values can significantly positively predict the professional development of primary and secondary school teachers (see the lines 118 to 119). Hypothesis 2(H2): Work engagement plays an mediating role between work values and the professional development of primary and secondary school teachers (see the lines 147 to 148). Hypothesis 3(H3): Perceived organizational support not only moderates the direct prediction of work values of professional development, but also moderates the prediction of work values on work engagement (see the lines 176 to 178). Moreover, every hypothesis has a plenty of evidences for inference.

4.Provide a brief theoretical background before presenting the research model. This background should succinctly summarize relevant theoretical frameworks or concepts that support the study's hypotheses and research model. This addition will enhance the theoretical foundation of the study and provide context for the research model.

My response: We have adopt your suggestion. we provide a brief theoretical background based on the Job Demands-Resources (JD-R) model and Social Exchange Theory (SET) before presenting the research model). You can see the changes from lines 179* to 187.

We find that the new added contents can indeed enhance the theoretical foundation of this research and provide context for the research model. We will be able to use the same way (your suggestion) when we write similar research articles in the future. We have got it. Thanks again! 

Methodology 

1.The methodology section appears to prioritize demographic details over discussing the actual study methodology. It is recommended to provide a brief discussion of the demographic characteristics within the text and present the detailed demographic information in a table for better readability and organization.

My response: Yes, just as you see, in the methodology section we describe more about demographic details, and we have adopt your advice: in order to have a better readability and organization, we have deleted some information and provide a brief discussion of the demographic characteristics. In addition, we have also added some contents (see the lines 201 to 207) to discuss the actual study methodology. You can see the changes from lines 201 to 212.

2.The methodology does not address how data were collected from multiple provinces. If the data were collected online, it raises questions about the high response rate of 95.7%, which is unusually high for online surveys. Authors should provide justification for this high response rate to ensure the reliability and validity of the collected data.

My response: We have added the information about how data were collected, and the added contents are as follow: the convenience sampling method was employed to select primary and secondary school teachers in Guangdong Province of China to complete the questionnaires online questionnaire platform Questionnaire Star (www.wjx.cn) from August 1 to October 30, 2023 (lines 201 to 207). The criteria for recruiting participants were primary and secondary school teachers who voluntarily wanted to participant should be recruited. The participants were informed of the anonymous submission of the questionnaires, study’s purpose, and confidentiality agreement. The participants could refuse or withdraw from the study anytime before submission. Just as you say, the data were collected online from questionnaire platform Questionnaire Star (www.wjx.cn), and the high response rate of 95.7%. There were some reasons as follow: First, We contacted the principals of primary and secondary schools where these teachers worked, and they all attached great importance to this matter and raised this matter at the school meeting, so that the teachers could actively cooperate. Second, we set up the questions in the questionnaire platform Questionnaire Star (www.wjx.cn) so that the participants could not submit if they did not compete all the questions. Third, The participants were informed of the anonymous submission of the questionnaires, study’s purpose, and confidentiality agreement. The participants could refuse or withdraw from the study anytime before submission, which gave them plenty of freedom and space. Last, we promised to give the results and conclusions of the this study back to the participants, and they were very careful to fill out the questionnaires.

3.The methodology discusses the measurement scales in detail. However, it would be more effective to consolidate this information into a single paragraph for clarity. Additionally, the discussion of reliability measures such as Cronbach's alpha and confirmatory factor analysis (CFA) should be moved to the results section, and the results should be presented in tabular format for easier reference.

My response: Thank you for your suggestion. We have consolidated the relevant information of measurement scales into a single paragraph for clarity. In addition, in the results section, we have added the discussion of reliability measures such as Cronbach's alpha and confirmatory factor analysis (CFA), and the results have been presented in tabular format for easier reference. You can see the changes from lines 214 to 233 (or see the table 1 or lines 257 to 274). 

4.As the authors is using control variables in the study also, I would suggest to provide brief detail about your control variables along with justification for using these variables as control.

My response: We added a part called “Control variables” in the article, and we controlled for demographic characteristics variables that may impact work values, work engagement, perceived organizational support, and professional development of primary and secondary school teachers, including gender, age,education level and teaching age (lines 235 to 237). For all regression analyses, we entered gender, age,education level and teaching age as control variables. All continuous variables were centered to avoid multicollinearity (lines 242 to 244). 

5.The track changes should be removed from the manuscript, particularly from lines 191 to 193, to ensure the document's professional presentation.

My response: Yes, we have removed all the track changes form the manuscript to make sure the document’s professional presentation (lines 196 to 199).

Results

1.The results section is commendably well-written and presented, demonstrating a strong presentation of findings. However, one aspect that requires attention is the discussion of scale reliability and validity. It is essential to include a detailed discussion of these aspects within the text of the results section, along with presenting the corresponding statistics in a table format. This will ensure a comprehensive evaluation of the measurement instruments used in the study and enhance the transparency of the findings.

My response: Thank you for your feedback and encouragement. In the results section, we have added a part called “Reliability and validity analysis”, and its contents includes Cronbach's alpha and confirmatory factor analysis (CFA). The results have been presented in a table format (see the table 1 or lines 256 to 273) for easier reference. Just as you say, this will ensure a comprehensive evaluation of the measurement instruments used in the study and enhance the transparency of findings. We have learned and in the following days wewill do like this when meeting the similar situation. Thanks again.

Discussion 

2.The discussion is again well written however I would suggest, Provide a more thorough integration of current results with previous research on work values, work engagement, and organizational support in the context of teacher professional development.

My response: Thank you for your feedback to the part of 

---

## [Decision Letter · Decision Letter 1]

26 Aug 2024

The impact of work values on the professional development of primary and secondary school teachers: A moderated mediation model

PONE-D-24-02012R1

Dear Dr. Shi,

We’re pleased to inform you that your manuscript has been judged scientifically suitable for publication and will be formally accepted for publication once it meets all outstanding technical requirements.

Kind regards,

Chunyu Zhang

Academic Editor

PLOS ONE

Additional Editor Comments (optional):

Reviewers' comments:

Reviewer's Responses to Questions

**Comments to the Author**

1. If the authors have adequately addressed your comments raised in a previous round of review and you feel that this manuscript is now acceptable for publication, you may indicate that here to bypass the “Comments to the Author” section, enter your conflict of interest statement in the “Confidential to Editor” section, and submit your "Accept" recommendation.

Reviewer #2: All comments have been addressed

2. Is the manuscript technically sound, and do the data support the conclusions?

Reviewer #2: Yes

3. Has the statistical analysis been performed appropriately and rigorously? 

Reviewer #2: Yes

4. Have the authors made all data underlying the findings in their manuscript fully available?

Reviewer #2: Yes

5. Is the manuscript presented in an intelligible fashion and written in standard English?

Reviewer #2: Yes

6. Review Comments to the Author

Reviewer #2: (No Response)

7. PLOS authors have the option to publish the peer review history of their article (what does this mean?). If published, this will include your full peer review and any attached files.

Reviewer #2: **Yes: **Sayed Farrukh Ahmed

---

## [Editor Report · Acceptance letter]

5 Sep 2024

PONE-D-24-02012R1 

PLOS ONE

Dear Dr. Shi, 

I'm pleased to inform you that your manuscript has been deemed suitable for publication in PLOS ONE. Congratulations! Your manuscript is now being handed over to our production team.

Kind regards, 

on behalf of

Dr. Chunyu Zhang 

Academic Editor

PLOS ONE